# Side-Arm Assisted Anilido-Imine Based Rare-Earth Metal Complexes for Isoprene Stereoselective Polymerization

**DOI:** 10.3390/molecules26144154

**Published:** 2021-07-08

**Authors:** Yi Wu, Xinli Liu, Dongmei Cui

**Affiliations:** 1State Key Laboratory of Polymer Physics and Chemistry, Changchun Institute of Applied Chemistry, Chinese Academy of Sciences, Changchun 130022, China; wuyi@ciac.ac.cn; 2Department of Polymer Science and Engineering, University of Science and Technology of China, Hefei 230026, China

**Keywords:** rare-earth metal catalyst, polyisoprene, stereoselectivity

## Abstract

Anilido-imine ligands *o*-C_6_H_4_(NHAr_1_)(CH=NAr_2_), in which Ar_1_ is 2,6-diisopropylbenzyl group and Ar_2_ contains fluorine (HL_1_) or methoxyl (HL_2_) group on ortho-position of phenyl substituent, were synthesized for constructing rare-earth metals based complexes of **1a**–**1c** (HL_1_ based Sc, Lu, Y) and **2a**–**2c** (HL_2_ based Sc, Lu, Y). Based on their NMR spectra and X-ray single-crystal structures, the side-arm group of -F and -OMe is identified to chelate to the corresponding central metal. The twisted angles between two planes formed by chelated heteroatoms (N, N, F for HL_1_ and N, N, O for HL_2_) are observed, in which the largest dihedral angle (53.3°) for HL_1_-Y and the smallest dihedral angle (44.32°) for HL_2_-Sc are detected. After being activated by Al*^i^*Bu_3_ and [Ph_3_C][B(C_6_F_5_)_4_], these catalysts showed great activity for isoprene polymerization. Bearing the same ligand HL_1_, smaller scandium based complex **1a** and middle size of lutetium based **1b** provided lower *cis*-1,4-selectivity (57.3% and 64.2%), larger yttrium complex **1c** displayed high *cis*-1,4-selectivity (84%). Chelating by crowded HL_2_, small size of scandium complex **2a** provided impressive *trans*-1,4-selectivity (93.0%), middle lutetium based **2b** displayed non-selectivity and larger yttrium complex **2c** showed clear *cis*-1,4-selectivity (83.3%). Moreover, **2a**/Al*^i^*Bu_3_ system showed the quasi-living chain transfer capability.

## 1. Introduction

According to the different regio- and stereo-selective polymerization method, polyisoprene is commonly divided into *cis*-1,4-polyisoprene, *trans*-1,4-polyisoprene and 3,4-polyisoprene. Among them, *cis*-1,4-polyisoprene has similar chemical composition, stereoselectivity and mechanical properties to natural rubber. It is widely used in tires, conveyor belts, adhesives, sports equipment, etc. In recent years, due to the shortage of natural resources and the aggravation of environmental pollution, it has become a trend to produce high performance tires with low rolling resistance, high wet skid resistance and low fuel consumption [1]. It is reported that the high-performance rubber can be obtained by mixing a small amount of *trans*-1,4-polyisoprene with *cis*-1,4-polyisoprene or natural rubber [2]. In addition, *trans*-1,4-selective polyisoprene has unique applications in medical materials and shape memory materials, etc. For example, *trans*-1,4-polyisoprene is an ideal material for making medical splints, orthopedic components and prosthetics.

So far, a large number of rare-earth metal catalysts for the *cis*-1,4-selective and 3,4-selective polyisoprene have been reported [3,4,5,6,7,8,9,10,11,12], such as aryldiimine (NCN)-ligated rare-earth metal dichlorides (*cis*-1,4-selectivity up to 98.8%) [13]; bis(carbene)phenyl (CCC) rare-earth metal dibromides (*cis*-1,4-selectivity up to 99.6%) [14]; bis(phosphino)-carbazolide (PNP)-chelated rare-earth metal complexes (*cis*-1,4-selectivity up to 99%) [15,16], etc. For 3,4-selective polyisoprene, there are amidino N-heterocyclic carbene ligated lutetium complex (3,4-selectivity 98.7%) [17], iminophosphonamido (NPN) ligated rare-earth metal bis(alkyl)s (3,4-selectivity 99.4%) and [Me_2_Si(C_5_Me_4_)(μ-PCy)YCH_2_SiMe_3_]_2_ (Cy = cyclohexyl) (3,4-selectivity 100%), etc. [18,19].

Comparatively, there are only a few reports on the *trans*-1,4-selective catalysts [2,20,21,22,23]. For the *trans*-1,4 polymerization of isoprene, the half-sandwich rare-earth metal complexes (CpQ)Ln(AlMe_4_)_2_ (CpQ = 2,3,4,5-tetramethyl-1-(8-quinolyl)cyclopentadienyl; Ln = Y, La) (*trans*-1,4-stereoselectivity >93%) [24], (Flut-Bu)La(AlMe_4_)_2_ (*trans*-1,4-stereoselectivity >85%) reported by Anwander and the chiral mononuclear dialkyl pincer complexes [(*S*,*S*)-BOPA]Ln(CH_2_SiMe_3_)_2_ (BOPA = (*S*,*S*)-bis(oxazolinylphenyl)amido; Ln = Sc, Lu) discovered by Xiaofang Li (*trans*-1,4-stereoselectivity > 99%) are efficient catalysts [25,26]. Interestingly, efficient modulating the stereoselective polymerization of isoprene has also been achieved by changing the size of metal center and/or rational adjusting the steric/electronic effect of chelating ligand or co-catalyst. For example, Shojiro Kaita et.al reported high *cis*-1,4/*trans*-1,4 selective polymerization of butadiene controlled by synergetic influence of metal size and type of AlR_3_ [27].

Previously we reported *β*-iminophosphonamine ligated complexes can realize the switchable stereoselectivity of polyisoprene from 3,4-selectivity of 94% to *trans*-1,4-selectivity of 95% by changing the metal size from Lu^3+^ (84.8 pm) to La^3+^ (106.1 pm) [2]. In addition to the type of metal and co-catalyst, the strategy of using the flexible side arm on ligand to adjust the electronic/steric properties of complex and thus finely change the capability of catalysts has also attracted much attention. As introduced by Yong Tang et.al, the additional “side-arm” groups near the catalytic center to regulate the electronic properties and spatial shape of the metal center, thus affecting the coordination of the monomer, and achieving the purpose of regulating the activity and catalytic properties of the catalyst [28,29]. However, there are few of report that fine modulate the stereoselectivity of catalyst by “side-arm strategy”.

Herein, we devised an aniline-imine ligated rare-earth metal complexes and introduced flexible heteroatom groups near the metal center to further control the polymerization process of isoprene. The skeleton of aniline-imine ligands might well be interpreted as a hybridization of classic salicylaldiminate and *β*-diketiminate ligands, which are widely applied to construct catalysts for olefin polymerization [30,31]. Through combining the “side-arm strategy” and suitable metal size influence, we have successfully realized the switchable stereoselectivity of polyisoprene from *cis*-1,4 to *trans*-1,4 selectivity. Furthermore, the efficient chain transfer polymerization for high *trans*-1,4 polymerization system was fine developed.

## 2. Results and Discussion

### 2.1. Synthesis and Characterization of Rare-Earth-Metal Complexes

The deprotonation of anilido-imine ligands with 1 equiv of rare-earth metal tris(alkyl)s afforded a series of rare-earth metal–bis(alkyl) complexes in high yields (Scheme 1). All the complexes are soluble in polar solvent like toluene and THF, but insoluble in unpolar hexane. The solid-state structures of complexes **1a**, **1c**, **2a** and **2c** were confirmed by X-ray diffraction measurements (Figure 1, Figure 2, Figure 3 and Figure 4). And the crystallographic data were summarized in Appendix A. Except for the coordinated bidentate ligands, the fluorine group and methoxy substituent on the phenyl group also coordinate to the corresponding central metal. The Y–N (amido) distances [2.313(5) Å for **1c**, 2.310(2) Å for **2c**] are shorter than the Y–N (imine) distances [2.451(5) Å for **1c**, 2.476(2) Å for **2c**], respectively. Except for **2a**, additional one THF molecule was involved in their crystal structures, which implies that **2a** owns the most crowded coordinating environment. A twisted angle between two planes formed by chelated heteroatoms (N, N, F for HL_1_ and N, N, O for HL_2_) were observed. Interestingly, the highest *cis*-1,4-selectivity of **1c** owns the largest dihedral angle (53.3°) among them, and the highest *trans*-1,4-selectivity of **2a** shows the smallest dihedral angle (44.32°), which are depicted below in detail. For complex **2a**, O, N1 and N2 all coordinate with the central metal scandium forming a chelating complex. The O–Sc bond length is 2.2387(13), Sc–N1 bond length is 2.1415(15), Sc–N2 bond length is 2.2540(15), which are close to the bond lengths between scandium and coordinated nitrogen atoms in the chelating catalyst [(*S*,*S*)-BOPA]Sc(CH_2_SiMe_3_)_2_ (BOPA = (*S*,*S*)-bis(oxazolinylphenyl)amido) reported by Xiaofang Li. Therefore, their catalytic behaviors are similar, and they both show high *trans*-1,4 selectivity for isoprene polymerization [26].

The ^1^H NMR spectrum of complex **1a** (Appendix A) was indicative of the formation of bis(alkyl) species and the resonance at 0.25 ppm is attributed to the methylene protons of scandium alkyl species. Compared with many other rare earth metal alkyl species, the resonance shifts down due to the stronger Lewis acidity of the Sc^3+^ ion [2]. Correspondingly, the methylene protons of Lu–CH_2_SiMe_3_ and Y–CH_2_SiMe_3_ in **1b** and **1c** appear at −0.75 and −0.51 ppm at higher field, respectively (Appendix A).

### 2.2. Polymerization of Isoprene

Isoprene polymerization was investigated in detail and the representative data are summarized in Table 1. The isoprene polymerization was initiated immediately as the complex was activated by [Ph_3_C][B(C_6_F_5_)_4_] and aluminum alkyls. The monomer was completely consumed within 0.5 h at room temperature (Table 1, entries 1–9).

Bearing the ligand L_1_, the scandium complex **1a** had medium *cis*-1,4-selectivity (57.3%), while yttrium complex **1c** and **2c** provided predominant *cis*-1,4-selectivity (84.0%, 83.3%) (Table 1, entries 1, 3, 7). This may be due to the fact that the larger yttrium provided a more open space, which is beneficial to the *η*^4^-*cis*-coordinating mode of monomer. Bearing the same metal center, the complex **2b** had medium *trans*-1,4-selectivity (58.3%), while the complex **1b** had medium *cis*-1,4-selectivity (64.2%) (Table 1, entries 2, 6). This may be attributed to the small steric hindrance of fluorine group facilitating *η*^4^-coordination. On the other hand, the electronic absorption of fluorine group increased the Lewis acidity of Lu^3+^ ion, which could increase the chance of *η*^4^-coordination [18]. Surprisingly, scandium complex **2a** displayed high *trans*-1,4-selectivity (93.0%), and exhibited a promising catalytic activity even at a low temperature (−30 °C, 98% yield in 4 h) with an increased *trans*-1,4-selectivity (97.0%) (Table 1, entries 4–5). Compared with **1a**, the side arm of ligand L_2_ adopted methoxy group with larger steric hindrance, making the ligand have a bulky space shielding on the metal center. The more planar structure of complex **2a** together with bulky space may only allow *η*^2^-*trans*-coordination mode of isoprene, on the other hand, the consecutive insertion of *trans*-monomer into the allyl–metal bond of active species with *syn*-prenyl moiety and *anti*-*syn* isomerization prior to monomer insertion lead to *trans*-1,4 unit [32,33,34,35,36,37].

It is well known that Al*^i^*Bu_3_ is not only used as a cocatalyst for the activation of catalyst precursors, but also as a chain transfer agent to regulate the molecular weight. In general, the molecular weight distribution increases with the amount of Al*^i^*Bu_3_ due to the deactivation of active sites by excessive Al*^i^*Bu_3_ [38,39,40]. In contrast, using the **2a**/Al*^i^*Bu_3_/[Ph_3_C][B(C_6_F_5_)_4_] system, the molecular weight of PIP decreased inversely with the increase of [Al]/[**2a**] ratio, meanwhile the molecular weight distribution were almost the same, showing the quasi-living chain transfer mode. For example, with the increase of [Al]/[**2a**] ratios from 2:1 to 40:1, the molecular weight of polyisoprene decreased from 4.58 × 10^4^ to 0.85 × 10^4^ g/mol, together with a slightly lower selectivity was observed (Table 1, entries 4, 8–11). 

## 3. Materials and Methods

### 3.1. General Information

All manipulations were performed under nitrogen atmosphere using standard high-vacuum Schlenk techniques or in a glovebox (Braun, Germany). All solvents were purified with a SPS system (Braun, Germany). The NMR data of the organometallic samples were obtained on a Bruker AV500 spectrometer (Switzerland) in chloroform-d or benzene-d6 at room temperature. The molecular weight and molecular weight distribution of the polyisoprene were measured with a HLC-8420 GPC (Tosoh corporation, Shunan, Yamaguchi, Japan) at 40 °C using THF as eluent (the flow rate was 0.35 mL/min) against polystyrene standards. Differential scanning calorimetry analyses were carried out on a Q 100 DSC (METTLER TOLEDO, Switzerland) from TA instrument under a nitrogen atmosphere. Any thermal history difference in the polymers was eliminated by first heating the specimen to above 80 °C, then cooling to −80 °C at 10 °C/min, and finally recording the second DSC scan from −80 to 80 °C at 10 °C/min. Isoprene was dried over CaH_2_ with stirring for 48 h and distilled under vacuum before use. [Ph_3_C][B(C_6_F_5_)_4_] was synthesized following the literature [41].

### 3.2. Synthesis

#### 3.2.1. Synthesis of *o*-C_6_H_4_NH(C_6_H_4_-F-o)(CH=NC_6_H_3_-i-Pr_2_-2,6) (L_1_)

Anilido-imine ligands L_1_–L_2_ were prepared according to the literature procedures [30,31]. Taking the synthesis method of ligand L_1_ as an example: *o*-fluorobenzaldehyde (6.2 g, 50 mmol), 2,6-diisopropylaniline (8.9 g, 50 mmol) and MgSO_4_ were mixed in hexane and stirred for 1 h. The mixture was filtered, and the solvent was removed to obtain the yellow solid *o*-C_6_H_4_F(CHNC_6_H_3_-*i*-Pr_2_-2,6), which was recrystallized in hexane to obtain pure product. A solution of n-BuLi (6 mL, 14.7 mmol) in hexane was added into hexane solution of *o*-fluoroaniline (1.4 mL, 14 mmol) at −78 °C, and the white lithium salt LiNHAr precipitated immediately, then the mixture was warmed to room temperature and stirred for 2 h, then it was added into a solution of *o*-C_6_H_4_F(CHNC_6_H_3_-*i*-Pr_2_-2,6) (4 g, 14 mmol) in THF at 25 °C. After stirring 5 h, the reaction was terminated by adding water, extracted with hexane and the solvent was removed to obtain the yellow solid crude product. The pure product was obtained by recrystallization in hexane (3 g, 60%). ^1^H NMR (500 MHz, CDCl_3_, 7.26 ppm, 25 °C): δ 1.19 (d, 12H, CH(CH_3_)_2_), 3.07 (m, 2H, CH(CH_3_)_2_), 6.88 (t, 1H, Ph-H), 7.03–7.4 (m, 9H, Ph-H), 7.56 (t, 1H, Ph-H), 8.33 (s, 1H, CH = NAr), 11.12 (s, 1H, NH). Analytical Calculated (Anal. Calcd) for C_25_H_27_FN_2_ (%): C, 80.18; H, 7.27; N, 7.48. Found: C, 80.45; H, 7.01; N, 7.73.

#### 3.2.2. Synthesis of *o*-C_6_H_4_NH(C_6_H_4_-OMe-o)(CH=NC_6_H_3_-i-Pr_2_-2,6) (L_2_)

The preparation method of ligand L_2_ is similar to that of ligand L_1_. ^1^H NMR (500 MHz, CDCl_3_, 7.26 ppm, 25 °C): δ 1.19 (d, 12H, CH(CH_3_)_2_), 3.10 (m, 2H, CH(CH_3_)_2_), 3.77 (s, 3H, OCH_3_), 6.82 (t, 1H, Ph-H), 6.92–6.98 (m, 2H, Ph-H), 7.04 (t, 1H, Ph-H), 7.12–7.31 (m, 4H, Ph-H), 7.37 (d, 1H, Ph-H), 7.44 (d, 1H, Ph-H), 7.56 (d, 1H, Ph-H), 8.3 (s, 1H, CH = NAr), 10.95 (s, 1H, NH). Anal. Calcd for C_26_H_30_N_2_O (%): C, 80.79; H, 7.82; N, 7.25. Found: C, 80.55; H, 8.08; N, 7.51.

#### 3.2.3. Synthesis of L_1_Sc(CH_2_SiMe_3_)_2_(THF) (1a)

The hexane solution (4.0 mL) of Sc(CH_2_SiMe_3_)_3_(THF)_2_ (0.22 g, 0.5 mmol) was added dropwise to the ligand L_1_ solution (0.187 g, 0.5 mmol in 4 mL hexane) at 0 °C. The mixture was stirred for 1 h and then cooling to −30 °C for 1 day afforded crystalline solids, which dried in vacuo to give orange solids of **1a** (0.23 g, 69%). Single crystals suitable for X-ray analysis were obtained from hexane at −30 °C. ^1^H NMR (500 MHz, C_6_D_6_, 7.16 ppm, 25 °C): δ 0.11 (s, 18H, CH_2_SiMe_3_), 0.25 (s, 4H, CH_2_SiMe_3_), 0.99 (d, 6H, CH(CH_3_)_2_), 1.24 (m, 4H, THF), 1.29 (d, 6H, CH(CH_3_)_2_), 3.31 (br s, 2H, CH(CH_3_)_2_), 3.59 (m, 4H, THF), 6.43–7.39 (m, 11H, Ph-H), 8.00 (s, 1H, CH = NAr). ^13^C NMR (125 MHz, C_6_D_6_, 128 ppm, 25 °C): 3.22, 25.6, 29.0, 45.9, 68.5, 114.9, 118.7, 119.5, 120.5, 121.8, 124.4, 126.5, 135.2, 136.2, 138.4, 141.5, 146.6, 151.8, 157.3, 159.1, 170.4. Anal. Calcd for C_40_H_64_FN_2_OScSi_2_ (%): C, 67.75; H, 9.10; N, 3.95. Found: C, 67.53; H, 8.90; N, 4.20.

#### 3.2.4. Synthesis of L_1_Lu(CH_2_SiMe_3_)_2_(THF) (1b)

The preparation method of complex **1b** is similar to that of complex **1a** (67%). ^1^H NMR (500 MHz, C_6_D_6_, 7.16 ppm, 25 °C): δ −0.77 (s, 4H, CH_2_SiMe_3_), 0.08 (s, 18H, CH_2_SiMe_3_), 0.92 (d, 6H, CH(CH_3_)_2_), 1.23 (m, 4H, THF), 1.34 (d, 6H, CH(CH_3_)_2_), 3.21 (m, 2H, CH(CH_3_)_2_), 3.85 (m, 4H, THF), 6.36–7.48 (m, 11H, Ph-H), 7.96 (s, 1H, CH = NAr). ^13^C NMR (125 MHz, C_6_D_6_, 128 ppm, 25 °C): 4.28, 25.0, 25.4, 28.9, 41.2, 70.3, 114.4, 115, 116, 120.5, 124.1, 125.6, 126.1, 127.2, 133.4, 134.8, 140.6, 141.5, 147.9, 149.2, 156.2, 158.1, 168.6. Anal. Calcd for C_40_H_64_FLuN_2_OSi_2_ (%): C, 57.26; H, 7.69; N, 3.34. Found: C, 57.56; H, 7.90; N, 3.08.

#### 3.2.5. Synthesis of L_1_Y(CH_2_SiMe_3_)_2_(THF) (1c)

The preparation method of complex **1c** is similar to that of complex **1a** (65%). Single crystal suitable for X-ray analysis was obtained from hexane at −30 °C. ^1^H NMR (500 MHz, C_6_D_6_, 7.16 ppm, 25 °C): δ −0.51 (s, 4H, CH_2_SiMe_3_), 0.11 (s, 18H, CH_2_SiMe_3_), 0.94 (d, 6H, CH(CH_3_)_2_), 1.22 (m, 4H, THF), 1.36 (d, 6H, CH(CH_3_)_2_), 3.21 (m, 2H, CH(CH_3_)_2_), 3.84 (m, 4H, THF), 6.38–7.49 (m, 11H, Ph-H), 7.98 (s, 1H, CH = NAr). ^13^C NMR (125 MHz, C_6_D_6_, 128 ppm, 25 °C): 4.13, 25.33, 28.95, 35.30, 70.30, 114.22, 115.45, 120.66, 124.15, 125.90, 126.07, 127.13, 133.29, 134.88, 140.45, 141.33, 147.73, 148.53, 156.32, 158.12, 168.18. Anal. Calcd for C_40_H_64_FN_2_OSi_2_Y (%): C, 63.80; H, 8.57; N, 3.72. Found: C, 63.59; H, 8.81; N, 3.95.

#### 3.2.6. Synthesis of L_2_Sc(CH_2_SiMe_3_)_2_ (2a)

The hexane solution (4 mL) of Sc(CH_2_SiMe_3_)_3_(THF)_2_ (0.22 g, 0.5 mmol) was added dropwise to the ligand L_2_ solution (0.193 g, 0.5 mmol in 4 mL hexane) at 0 ℃. The mixture was stirred for 1 h and then cooling to −30 °C for 1 day afforded crystalline solids, which dried in vacuo to give orange solids of **2a** (0.22 g, 73%). ^1^H NMR (500 MHz, C_6_D_6_, 7.16 ppm, 25 °C): δ −0.05 (s, 4H, CH_2_SiMe_3_), 0.00 (s, 18H, CH_2_SiMe_3_), 1.01–1.33 (m, 12H, CH(CH_3_)_2_), 3.37 (m, 2H, CH(CH_3_)_2_), 3.90 (s, 3H, OCH3), 6.48–7.45 (m, 11H, Ph-H), 8.07 (s, 1H, CH = NAr). ^13^C NMR (125 MHz, C_6_D_6_, 128 ppm, 25 °C): 3.25, 26.07, 58.54, 111.28, 117.43, 118.51, 120.46, 120.99, 121.24, 124.14, 124.44, 135.15, 136.36, 139.96, 141.81, 147.12, 149.88, 153.11, 170.63. Anal. Calcd for C_37_H_60_N_2_OScSi_2_ (%): C, 68.37; H, 9.30; N, 4.31. Found: C, 68.72; H, 9.04; N, 4.58.

#### 3.2.7. Synthesis of L_2_Lu(CH_2_SiMe_3_)_2_(THF) (2b)

The preparation method of complex **2b** is similar to that of complex **2a** (70%). ^1^H NMR (500 MHz, C_6_D_6_, 7.16 ppm, 25 °C): δ −0.59, −0.84 (4H, CH_2_SiMe_3_), 0.00 (s, 18H, CH_2_SiMe_3_), 0.96–1.24 (m, 12H, CH(CH_3_)_2_), 1.37 (m, 4H, THF), 3.22 (m, 2H, CH(CH_3_)_2_), 3.55 (m, 4H, THF), 3.75 (s, 3H, OCH_3_), 6.44–7.45 (m, 11H, Ph-H), 7.94 (s, 1H, CH=NAr). ^13^C NMR (125 MHz, C_6_D_6_, 128 ppm, 25 °C): 3.88, 25.71, 29.06, 58.01, 67.99, 111.60, 117.28, 120.12, 120.45, 120.62, 121.47, 124.43, 135.06, 136.57, 140.90, 141.64, 145.97, 149.98, 154.17, 171.05. Anal. Calcd for C_41_H_67_LuN_2_O_2_Si_2_ (%): C, 57.86; H, 7.93; N, 3.29. Found: C, 57.63; H, 8.14; N, 3.53.

#### 3.2.8. Synthesis of L_2_Y(CH_2_SiMe_3_)_2_(THF) (2c)

The preparation method of complex **2c** is similar to that of complex **2a** (74%). Single crystals suitable for X-ray analysis were obtained from hexane at −30 °C. ^1^H NMR (500 MHz, C_6_D_6_, 7.16 ppm, 25 °C): δ −0.46, −0.68 (4H, CH_2_SiMe_3_), 0.06 (s, 18H, CH_2_SiMe_3_), 1.00–1.33 (m, 12H, CH(CH_3_)_2_), 1.37 (m, 4H, THF), 3.19 (m, 2H, CH(CH_3_)_2_), 3.61 (m, 4H, THF), 3.78 (s, 3H, OCH_3_), 6.52–7.63 (m, 11H, Ph-H), 7.94 (s, 1H, CH = NAr). ^13^C NMR (125 MHz, C_6_D_6_, 128 ppm, 25 °C): 4.21, 25.38, 29.00, 31.97, 34.57, 34.88, 57.66, 69.37, 110.62, 116.03, 117.53, 119.03, 123.94, 124.19, 124.50, 127.20, 128.35, 133.77, 135.46, 140.82, 142.63, 147.58, 150.02, 151.29, 168.84. Anal. Calcd for C_41_H_67_N_2_O_2_Si_2_Y (%): C, 64.37; H, 8.83; N, 3.66. Found: C, 64.65; H, 8.59; N, 3.96.

### 3.3. Polymerization of Isoprene

In a glovebox, a toluene solution (3 mL) of complex **1a** (6.6 mg, 10 μmol), 1 equiv. of [Ph_3_C][B(C_6_F_5_)_4_] (9.2 mg, 10 μmol) and 0.1 mmol of Al*^i^*Bu_3_ were added to a 10 mL flask. The toluene solution of isoprene (1 mL of toluene, 10 mmol, 0.68 g) was added to the catalytic system and stirred for 0.5 h. Then, the polymerization was terminated by injecting methanol. The mixture was poured into methanol containing an appropriate amount of BHT to precipitate the products. The obtained polymer was collected by filtration, washed with methanol, and dried to constant weight in vacuum at 35 °C. Then the conversion can be calculated from the mass of the polymer (Table 1, Entry 1).

### 3.4. X-ray Crystallographic Studies

Data collections of Single crystals of complexes were performed at −100 °C on a Bruker SMART APEX diffractometer with a CCD area detector, using graphite-monochromated Mo Kα radiation (λ = 0.71073 Å). The SMART program package was used to determine the unit-cell parameters. The absorption correction was applied using SADABS. The structures were solved by direct methods and refined on *F*^2^ by full-matrix least-squares techniques with anisotropic thermal parameters for non-hydrogen atoms. Hydrogen atoms were placed at calculated positions and were included in the structure calculation without further refinement of the parameters. All calculations were carried out using the Olex 2 program. Molecular structures were generated using the ORTEP program.

## 4. Conclusions

In summary, a series of rare-earth metal complexes (Ln = Sc, Lu, Y) bearing anilido-imine ligand were successfully synthesized and defined, which were activated with [Ph_3_C][B(C_6_F_5_)_4_] and aluminum alkyls to initiate polymerization of isoprene. The choice of central metal size and the introduction of heteroatom coordination groups with different steric hindrance on the side arm of the ligand are the key to adjust the stereoselectivity of polyisoprene. Therefore, by adjusting these two factors, the stereoselectivity of polyisoprene was successfully switched from *cis*-1,4 (84%) to *trans*-1,4 (93%). In addition, the quasi-living chain transfer polymerization for high *trans*-1,4 polyisoprene system was developed.

## Data Availability

All data is available in the main text, in the Appendix A, or on reasonable request.

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
