# Peer review of "Side-Arm Assisted Anilido-Imine Based Rare-Earth Metal Complexes for Isoprene Stereoselective Polymerization"

_molecules, 2021, doi:10.3390/molecules26144154_

Round 1

Reviewer 1 Report

This study deals with new (NN) rare-earth based catalysts for the polymerization of isoprene.

Some new complexes are described, which are further assessed as pre-catalysts for isoprene polymerization. The catalytic results are interesting, especially showing a rare occurrence of trans-1,4 selectivity with a scandium catalyst. However, this work suffers from important weaknesses, which lead me to ask for major changes to be made in order to accept it.

I noticed that no elemental analysis (or eventually HR-MS) is given for the L1 and L2 ligands, and not for all organometallic complexes either, which may appear incomplete for a high-standard journal like “Molecules”

I notice also that no BHT is added in the polymer medium before precipitation (experimental), I guess this is an omission in the description of the experimental part.

In the introductive/state of the art section, one former paper by Visseaux and coworkers should be cited, where a scandium catalyst activated by borate/alkylAl combination affords trans-selectivity of 94.1%. (Dalton Trans, 2013, 42, 790). Besides, it is unseemly to ignore the Nd/Mg catalytic system of the same author which leads to 98.5% 1,4 trans PI (Macromolecules 2005, 38, 3162)

Regarding the polymerization part: the authors speak about deactivation when AlR3 is added as CTA in excess in the reaction mixture.  I consider that this may not systematically take place. Indeed, the reference cited by the authors illustrates the exact contrast to what they claim: from 10 to 70 equiv. Al-CTA the MWD remains constant (1.36-1.33). Please reconsider the discussion on that point.

About the polymerization mechanism: one important paper by Luo and coworkers (Macromolecules, 2014, 10.1021/ma500988s, not cited) strongly suggests, on the basis of a convincing DFT approach, that the trans-1,4-selectivity is not simply resulting from a single-coordination h2 of the diene monomer to the metal. From this paper, the single-coordination advanced by the authors (page 5, bottom: “which is the key to form high trans-1,4…”) in the present article does not seem to be sufficiently well founded to explain the trans-selectivity. See also the widely accepted mechanism of Porri regarding conjugated dienes coordination polymerization (Progr. Polym. Sci. 1991, 16, 405) later generalized by Tobisch (Can J. Chem, 2009, 87, 1392).

Finally, it seems to me that the role played by the fluorine-substituted ligand is not well discussed in term of electronic effect (steric effects are considered mainly) toward the polymerization activity/selectivity.

Miscellaneous:

  • angle and not “angel” in the abstract (three occasions)
  • page 1, bottom: selective and not “selectivie”
  • references : ref 2, 2015 in bold; ref 16, isoprene polymerization (space added)

Author Response

Response to Reviewer 1 Comments

Point 1: I noticed that no elemental analysis (or eventually HR-MS) is given for the L1 and L2 ligands, and not for all organometallic complexes either, which may appear incomplete for a high-standard journal like “Molecules”. 

Response 1: Thanks for your kind remind. Elemental analysis for the ligands L1, L2 and all organometallic complexes has been added.

Point 2: I notice also that no BHT is added in the polymer medium before precipitation (experimental), I guess this is an omission in the description of the experimental part. 

Response 2: Thanks for your remind. Indeed, BHT was added before precipitation, and the corresponding description was revised in the experimental part.

 Point 3: In the introductive/state of the art section, one former paper by Visseaux and coworkers should be cited, where a scandium catalyst activated by borate/alkylAl combination affords trans-selectivity of 94.1%. (Dalton Trans, 2013, 42, 790). Besides, it is unseemly to ignore the Nd/Mg catalytic system of the same author which leads to 98.5% 1,4 trans PI (Macromolecules 2005, 38, 3162). Response 3: Thanks a lot for your kind remind. The related works of Visseaux and coworkers are very important, and the two references are added into main text with proper introduction (Ref 16(c-d)). Point 4: Regarding the polymerization part: the authors speak about deactivation when AlR3 is added as CTA in excess in the reaction mixture. I consider that this may not systematically take place. Indeed, the reference cited by the authors illustrates the exact contrast to what they claim: from 10 to 70 equiv. Al-CTA the MWD remains constant (1.36-1.33). Please reconsider the discussion on that point. 

Response 4: Appreciate a lot for your comment. We have systematically examined the corresponding polymerization, and the obtained results are similar with the previous conclusion. Indeed, the reference (Ref. 26(a)) cited in this manuscript reported the results contrary to our claims, but the author of the reference mentioned “broadening molecular weight distribution is always concomitant as the active sites are deactivated or even poisoned by the presence of an excess amount of AlR3”. For example, the added two references (Ref. 26(b-c)) showed the deactivation when AlR3 is added as CTA in excess in the reaction mixture.

 Point 5: About the polymerization mechanism: one important paper by Luo and coworkers (Macromolecules, 2014, 10.1021/ma500988s, not cited) strongly suggests, on the basis of a convincing DFT approach, that the trans-1,4-selectivity is not simply resulting from a single-coordination η2 of the diene monomer to the metal. From this paper, the single-coordination advanced by the authors (page 5, bottom: “which is the key to form high trans-1,4…”) in the present article does not seem to be sufficiently well founded to explain the trans-selectivity. See also the widely accepted mechanism of Porri regarding conjugated dienes coordination polymerization (Progr. Polym. Sci. 1991, 16, 405) later generalized by Tobisch (Can J. Chem, 2009, 87, 1392). Response 5: Thanks a lot for your question. And yes, you are right. We realize that although the coordination mode of diene monomer to the metal affects the microstructure of polymer to a great extent, the stereoselectivity of polymer is finally determined by the following step of monomer inserting into the chain end. The consecutive insertion of trans-monomer into the allyl−metal bond of active species with syn-prenyl moiety and anti-syn isomerization prior to monomer insertion will lead to trans-1,4 unit. The relevant content has been revised in the manuscript. Also the references you mentioned have been cited (Ref. 24(b-d)). Thanks again for your comment. Point 6: Finally, it seems to me that the role played by the fluorine-substituted ligand is not well discussed in term of electronic effect (steric effects are considered mainly) toward the polymerization activity/selectivity. Response 6: Thank you for your question. In this manuscript, fluorine group and methoxy group are selected as coordination heteroatoms. Although their electronic effects are different, the monomer conversion can reach almost 100% in half an hour, and there is little difference in polymerization activity, so it is not discussed emphatically. As for the effect on the selectivity of polymerization, according to the reference (Ref. 15(a)), we can speculate that the weak electron-withdrawing fluorine group increases the Lewis acid of metal ion, which may increase the chance of isoprene η4-coordination, resulting in the increase of cis-1,4 content. The relevant content has been supplemented in the manuscript. Thanks a lot for your mind again. Point 7: Miscellaneous:•             angle and not “angel” in the abstract (three occasions)•             page 1, bottom: selective and not “selectivie”•             references : ref 2, 2015 in bold; ref 16, isoprene polymerization (space added) 

Response 7: Thanks for your kind remind. Spelling errors and references format errors have been corrected.

Reviewer 2 Report

This contribution by the Cui group is a nice piece of organometallic chemistry combined with an interesting polymerization study. The authors report on the synthesis of rare-earth metal complexes coordinated by anilido-imine ligands and examine the effect of the steric bulk/coordination of the side arm and ionic radii of the rare-earth metals on the performance in isoprene polymerization. Moreover, the effect of the chain transfer agent AliBu3 is nicely exemplified by variation of its concentration.

Please report the probability level of the X-ray structures (Figure 1-4) and the selectivity of PIP with decimals (Table 1, entry 3, 5, 11).

Wording/typos:

P1, Line 16-17: “angle (53.3°)” and not “angel (53.3 A)”  Line 21 crowded and not crowed

P6, Line 167: “benzene-d6” and not “benzene-d”

Exp. Part: Please add elemental analyses and IR data to the synthesis of compounds 1a-c and 2a-c.

Supporting Information: Please indicate residual solvent signals and impurities in the NMR spectra of 1a, 2a-c.

References:

- why not citing review articles on this topic? there is also a nice by one of the authors

- introduction: what do authors mean by …. There are only a few reports on the trans-1,4-polmerization due to the lack of efficient catalysts….? Are authors refereeing to catalyst activity or selectivity? There are catalyst systems from Visseaux and Anwander, affording polyisoprene microstructures of >98% trans-1,4. Therefore, I would not state that the 93% trans-1,4 obtained by the authors are impressive.

- the effect of sidearms in half-sandwich complexes has been demonstrated by others.

Author Response

Response to Reviewer 2 Comments

Point 1: Please report the probability level of the X-ray structures (Figure 1-4) and the selectivity of PIP with decimals (Table 1, entry 3, 5, 11).

Response 1: Thank you for your kind remind. X-ray structure of complexes with 35% probability thermal ellipsoids, which has been added to the manuscript. The data on the selectivity of polyisoprene in Table 1 are kept to one decimal place, which has been corrected in the manuscript.

Point 2: Wording/typos:

P1, Line 16-17: “angle (53.3°)” and not “angel (53.3 A)”  Line 21 crowded and not crowed

P6, Line 167: “benzene-d6” and not “benzene-d”

Response 2: Thanks for your remind. “angel (53.3 A)” has been amended to “angle (53.3°)”. “crowed” has been amended to “crowded”. “benzene-d” has been amended to “benzene-d6”.

Point 3: Exp. Part: Please add elemental analyses and IR data to the synthesis of compounds 1a-c and 2a-c.

Response 3: Thanks for your kind remind. Elemental analysis for all complexes has been added. As for IR data, I am so sorry that there is no suitable condition to measure the infrared analysis in the absence of water and oxygen as our compounds are not stable in air. We are sorry for that.

Point 4: Supporting Information: Please indicate residual solvent signals and impurities in the NMR spectra of 1a, 2a-c.

Response 4: Thanks a lot for your kind remind. Residual solvent signals and impurities in the NMR spectra of 1a, 2a-c have been marked in supporting information.

Point 5: References: why not citing review articles on this topic? there is also a nice by one of the authors

Response 5: Thanks for your kind remind. We have cited review articles on this topic. (Ref. 13(b), 25(b))

Point 6: introduction: what do authors mean by …. There are only a few reports on the trans-1,4-polmerization due to the lack of efficient catalysts….? Are authors refereeing to catalyst activity or selectivity? There are catalyst systems from Visseaux and Anwander, affording polyisoprene microstructures of >98% trans-1,4. Therefore, I would not state that the 93% trans-1,4 obtained by the authors are impressive.

Response 6: Thanks for your question. I mean, the trans-1,4-selective catalysts have still remained relatively scarce compared to the cis-1,4 catalysts, since their specific sterics and electronics demand the coordination/insertion modes of the monomers to the metal active species ( from Ref. 2: Organometallics 2015, 34, 4063–4068). The sentence your mentioned in introduction is not clear and has been revised to “there are only a few reports on the trans-1,4-selective catalysts”.

On the other hand, the highlight of the article is to regulate the polymerization behavior through the side arm effect. And the experimental results showed that the switch of stereoselectivity was achieved by regulating the side arm. Compared with the high trans-1,4 selectivity, the switched 93% trans-1,4 was interesting. Thanks again.

Point 7: the effect of sidearms in half-sandwich complexes has been demonstrated by others.

Response 7: Thanks for your kind remind. Tang Yong proposed "side-arm strategy" for the first time to guide the design and synthesis of ligands and catalysts, and realized the regulation of ethylene polymerization activity and stereoselectivity of styrene polymerization. “Side-arm strategy” has become a novel concept in catalyst design, which has been recognized and widely used by peers at home and abroad. We know that the effect of sidearms has been demonstrated by others. However, the stereoselectivity of conjugated diene polymerization controlled by “side-arm strategy” is rarely reported. Thanks for your comment again!

Round 2

Reviewer 1 Report

The manuscript has been improved appropriately to my point of view (just a small typo : "seleectivity ", page 5, line 141)

Author Response

Response to Reviewer 1 Comments

Point 1: The manuscript has been improved appropriately to my point of view (just a small typo : "seleectivity ", page 5, line 141) 

Response 1: Thanks for your kind remind. “seleectivity” has been amended to “selectivity”.

Reviewer 2 Report

Authors have addressed most of my criticism carefully. However, a new sentence has been added, which is pretty much incomprehensible. On page 5 … the electronic absorption of fluorine group increased the Lewis acidity???? There seems to be also a major formatting problem with Journal names.

Otherwise, no further objections to publication of this work in Molecules. Nice work.

Author Response

Response to Reviewer 2 Comments

Point 1: Authors have addressed most of my criticism carefully. However, a new sentence has been added, which is pretty much incomprehensible. On page 5 … the electronic absorption of fluorine group increased the Lewis acidity???? There seems to be also a major formatting problem with Journal names.

Response 1: Thanks for your question. Fluorine group is an electron withdrawing group, which can increase the electropositivity of metal center, in other words, increase the Lewis acid of metal center. And formatting problem has been amended. Thanks for your comments again.